# Ecological Network Theory Boosts Land Maxing Benefits for Biodiversity: An Example with Tropical Bee-Plant Interactions

**DOI:** 10.3390/insects16121269

**Published:** 2025-12-13

**Authors:** Valerie E. Peters, Elijah Cruz Cardona

**Affiliations:** 1Department of Biological Sciences, Eastern Kentucky University, 521 Lancaster Ave, Richmond, KY 40475, USA; 2Department of Entomology, University of Kentucky, Lexington, KY 40546, USA; e.cruz.cardona@uky.edu

**Keywords:** agroforestry, pollination services, modularity, centrality index, model selection

## Abstract

Conservation initiatives that simultaneously target biodiversity conservation and human livelihoods are a critical goal of ecological science, and one such approach involves scientifically guiding how agroforests should be managed so that they maximize their potential for supporting both people and wildlife. In this paper, we demonstrate the use of ecological network science to provide recommendations to farmers on how to manage diverse agroforests to maximize their potential to support wildlife, using bees as the example wildlife group. We compared results from two assemblages of the bee community, the entire bee community which included the economically important meliponine or stingless bees, and a subset of the entire bee community that included only the meliponine bees, to determine if network science derived agroforestry management practices would be similar between the two groups. We quantitatively identified key plant species for both bee communities, and identified which key plant species were shared between the two bee communities.

## 1. Introduction

The successful mitigation of climate change and deforestation impacts to biodiversity will require the human construction of novel, biodiverse, multi-use habitat types, in addition to the protection of natural areas [1,2]. One example of these multi-use habitat types, coined ‘land maxing’ in [3], includes, among other practices, the careful selection of a diverse assemblage of plants that can support higher numbers of animal species and also provide landowners with additional sources of income via both (a) the re-development of non-timber marketable traditional products and (b) the development of new, non-timber, niche market products [1]. Land maxing expands other ‘land sharing’ solutions, such as certified diverse agroforestry systems, where at least 12–15 tree species are included within each hectare of the production area, among other science-based sustainability standards (e.g., Smithsonian Migratory Bird Center Bird-Friendly Coffee). In land maxing each tree or shrub species incorporated into the agroforestry system will themselves produce or contribute to the production of non-timber marketable products, contribute highly nutritive forage for livestock, have the biggest impact on ecological network structure and stability, or possibly some combination of these. The development of non-timber niche market and marketable traditional products within diverse, multifunctional agroforestry systems is urgently needed in the global south to create new local business opportunities, promote social justice, halt biodiversity loss and environmental degradation, mitigate climate change impacts and kick-start rural economies [1,4].

One relatively underexplored non-timber marketable product with high potential for inclusion in land maxing agroforestry systems are the products derived from the stingless bees (tribe Meliponini). Meliponine bees are pantropical in distribution with >600 species worldwide [5]. Following the pattern of many mutualist species, stingless bees are most diverse in the Neotropics [6,7], which is home to over 80% of the species [5]. Stingless bees are important economically because of their effectiveness in pollination of tropical cash crops [8,9], medicinal honey production, and wax/resin production, as well as for the potential to develop novel, niche marketable products from their wax/resin/honey [10,11,12,13]. Stingless bees represent the largest biomass of bees in Neotropical wild bee communities, and because of their eusociality habit enabling them to remain active during the entire year [5], this proportion is even greater during months when other solitary and less social bee species are inactive or less abundant in the community [14]. Owing to their high biomass and their dominance on tropical trees and other blooms [5], stingless bees may play a key role in pollination services in tropical biodiversity hotspots where maintaining plant diversity is globally critical [15,16,17]. Despite their incredible importance, stingless bee populations are likely experiencing a similar trend to that of all wild bee populations; one of population decline [18,19,20]. Pesticide use in the Neotropics harms pollinators as they visit plants that have been sprayed [21,22]. Land-use change, and deforestation also has significant negative impacts on wild pollinator populations which includes stingless bees [9,23,24,25,26,27]. Finally, Neotropical bees may be highly vulnerable to climate change owing to their restrictive altitudinal distributions [14], their participation in mutualisms that are susceptible to phenological mismatches [28], and a reduction in nutrition available for bees due to declines in plant health [20,29].

The use of ecological network research in conservation science and practice is a developing and critical discipline of ecology [30,31]. Ecological networks can be characterized by various metrics or indices, and some of these values are posited to hold relationships with the conservation value of the network. For example, connectance, or the proportion of realized interactions out of those possible, is a measure of network complexity, and communities with high connectance are suggested to be more stable [32]. However, networks with high modularity, or subsets of species that interact more frequently with each other than with other species outside the module, and high network specialization, have also been posited to promote greater network stability and thus have higher conservation value [32,33,34,35,36,37] Finally, the network metric, nestedness, which shows subsets of specialized interactions nested within generalist interactions, has been shown to increase the conservation value of the ecological network through protecting the network from species loss [30]. To improve the value of ecological network theory for conservation, more datasets from underrepresented biomes and taxonomic groups such as tropical bee communities are needed to understand which of these metrics should be the target of conservation efforts. For bee communities, pristine and stable ecological networks are proposed to be synonymous with protecting pollination services, although much work remains to verify how network structure influences ecosystem processes and ecosystem functioning [37,38,39,40].

The goal of our study was to quantify bee-plant interactions of tropical agroforests to provide some guidelines or management recommendations to farmers that wish to maximize the potential of their already diverse agroforests to further protect bee-plant interactions. Within this, we focused on two objectives. The first was to understand how bee species richness, bee abundance, bee community composition, and the network indices of connectance, modularity, specialization and nestedness are related to agroforestry management practices of diverse agroforests. The second objective was to quantify which plant species are key in supporting structure and stability of the bee-plant interaction networks. Finally, we compared results between the entire bee community, as well as a subset of the entire bee community that included only the economically important stingless bees, to determine if management recommendations would be similar between the two groups. This question was motivated by the idea that farmers could more easily monitor or quantify the stingless bee community in their agroforests, and may also have more incentive for protecting this particular subset of the bee community for their economic value.

## 2. Materials and Methods

### 2.1. Study Area

The research sites included 10 agroforests in San Luis de Monteverde, Costa Rica (Figure 1). Agroforestry systems are land management practices that integrate trees and shrubs with crops and/or livestock, creating environmental, economic, and social benefits. Five were primarily silvopastoral agroforestry systems, defined as agroforests where the understory is cultivated for cattle grazing, and the other five were coffee agroforestry systems, where coffee plants are cultivated in the understory. Both coffee agroforests and silvopasture agroforests were on average one-hectare in size. All 10 agroforests were located between 800 and 1000 m elevation, embedded in a landscape typical of the tropical countryside, with forest patches and riparian forest adjacent to or in close proximity. Agroforests were selected for this study only if they had a diverse assemblage of trees within the production area (see Table 1 for detailed description of the number of trees and number of tree species within each agroforest), and landowners provided permission. Common trees shared by the two agroforestry included *Psidium guayaba*, *Musa* spp., *Citrus* spp., *Persea americana*, *Daphnopsis americana*, *Montonoa guatamelensis*, *Croton niveus*, *Bursera simarouba*, and *Sapium glandulosum*. All agroforests were separated by at least 200 m. Research permit was obtained from Costa Rica National System of Conservation Areas (SINAC).

### 2.2. Data Collection

Each of the 10 agroforests was sampled for flowering plants and bees three times during the rainy season from 5 June–24 July and once during the dry season from 7–19 December 2022. Measurements of agroforest vegetation structural variables were conducted in June 2022 (Appendix A).

#### 2.2.1. Bee Sampling

Bees were collected to obtain two variables of the bee communities in each agroforest: (1) bee species richness, and (2) bee abundance. During each of the four sampling periods, we used the method of discriminate sweeping to collect all bees observed to land on open flowers [41]. Each sampling period consisted of three observers conducting an exhaustive survey of the entire one-ha agroforest, searching for open flowers on each tree, shrub, and herbaceous plant species from 8:00–14:00 h (CST) except during intermittent short periods of heavy rain. Once a flowering plant was located, it was observed for bee activity for at least 30-min. If other individuals of that plant species with open flowers were located in the agroforest, subsequent observations of lesser time were conducted at different individuals. Plant species were observed for bees during each sampling period that the plant species had open flowers. Once a bee was observed to land on a flower, a hand-held insect sweep net (Lito Enterprise Society, Dongshan Township, Taiwan) or plastic collecting jar (Bioquip, Rancho Dominguez, CA, USA) was used to collect the individual. Only one attempt was made to collect each bee that landed on a flower, as all missed attempts would result in the bee flying away. If a plant species’ open flowers were all above 8 m, bee sampling was not conducted, although this only occurred for one tree species in one of the agroforests.

Bees were also collected by setting up two honey-bait stations in each agroforest during each of the four sampling periods. Honey-bait stations were comprised of a 3:1 water:honey mixture that was sprayed at the beginning of each sampling period on vegetation without flowers, with one station located close to a bordering forest patch and a second located near the center of the agroforest [42]. Bees were then collected from each station for two 30-min observations, with one sampling period conducted between 10:00–11:30 h, and the second conducted between 12:00–14:00 h. Honey-bait stations were used to contribute to the quantifying of stingless bee species richness in agroforests only, as this method is not recommended to quantify stingless bee abundance owing to their eusociality habit. Therefore, our variable of bee species richness was determined by combining honey-bait station data with floral visitation data, whereas only bees collected directly from flowers were used for the calculation of bee abundance and in bee-plant interaction networks (described below).

All collected bees were preserved in ethanol and returned to Eastern Kentucky University for processing and identification. Megachilidae, ceratinine and augochlorine bees were identified to species or morphospecies using a variety of references [43,44], keys, and a reference collection housed at Eastern Kentucky University. Meliponine bees were keyed to species following [45] and keys from University of Costa Rica faculty. Tapinotaspidine bees were keyed to species or morphospecies following [46]. All specimens are housed in the Peters lab at Eastern Kentucky University.

#### 2.2.2. Plant Sampling

During each of the four sampling periods an exhaustive survey of all flowering plants was conducted within the entire one-ha area of the agroforest. All flowering plants from which bees were collected were identified to species or morphospecies in the field using iNaturalist, local expert help and field guides [47,48]. In addition, the total number of open blooms on each individual plant was counted during each sampling period.

Agroforest vegetation structural variables were quantified in June 2022. Tree species richness was estimated by identifying all tree species within the one-ha agroforest to species or morphospecies in the field using iNaturalist, local expert help and field guides [47,48]. Tree size was determined by two measurements: (1) the trunk diameter at breast height (DBH) using a DBH tape, and (2) tree height, using a rangefinder. We used the standard geometric formula of diameter = circumference/π that is calibrated on the DBH tape itself. Percent shade cover of each one-ha agroforest was quantified along two 100-m transects that crossed in the middle of the farm, using a GRS densitometer (Table 1 and Appendix A).

### 2.3. Statistical Analysis

To understand how bee species richness, bee abundance and the network indices of connectance, modularity, specialization, and nestedness were related to agroforestry practices in diverse agroforests, we used Generalized Linear Mixed Models (GLMM) and linear regression models. GLMMs with a negative binomial error distribution were used for bee species richness and abundance by sampling period, with the agroforest ID as the random term. Network indices were obtained from bee-plant interactions across three sampling periods, and therefore linear regression models were used for these response variables. Network indices were log transformed to meet the assumptions of normality. Negative binomial and normal error distribution fit for each response variable were evaluated using a chi-square goodness-of-fit test.

Using the “AICcmodavg” package in R [49] we then performed AICc model selection of a candidate set of models that included the following fixed effects in single predictor models: (1) percent shade cover, (2) the number of flowering plant species, (3) the number of tree species, (4) total tree DBH, (5) average tree height, (6) average tree DBH, (7) the total number of open blooms, (8) the total number of trees. Two additional models were also included in the candidate set: a null model that included either the intercept only or the random term and the intercept but no fixed effects, and a model that included the effect of the number of tree species and the number of flowering plant species. The R^2^ was then calculated for the most plausible models, using the “MUMIn” package in R [50] for the GLMMs to enable reporting of both the fixed effects only (marginal R^2^) and the fixed and random effects together (conditional R^2^).

We also directly compared bee species richness and abundance between the two agroforestry types using rarefaction to compare bee species richness, and a likelihood ratio test comparing two nested GLMM models, one with only the intercept and the random effect of agroforest ID and one with the fixed effect of agroforest type and the random term agroforest ID. A species accumulation curve was constructed to assess the completeness of our sampling effort.

All statistical analyses were conducted in R version 4.0.3 [51].

#### 2.3.1. Network Indices

Network indices were estimated from each agroforest separately. Bee-plant interaction networks were of two types for each agroforest: (1) the entire bee community and (2) the subset community of meliponine bees. Both network types were quantitative, or abundance-based, and included the individual agroforest interactions across all three sampling periods conducted during the rainy season. Dry season interaction networks were not constructed for each individual agroforest due to the small sample size of only one sampling period. All network indices were calculated using the “bipartite” package in R [52]. The following four network indices were selected:(1)The weighted connectance (w-C) algorithm was used to obtain a measure of network connectance, or the sum of all realized links in a network divided by the possible links [53]. Its values range between 0 and 1, with higher values indicating increases in realized interactions.(2)The networklevel modularity function was used to estimate modularity for each network. Modularity values range from 0 to 1, where higher values indicate more subsets of species interacting more among themselves, as compared to other species in the network [54](3)H2′ was used to quantify specialization in each network [55]. Its value characterizes the average degree of specialization between the species in the entire network. H2′ values range from 0 to 1, with higher values indicating higher network specialization.(4)The weighted nestedness metric based on overlap and decreasing fill (w-NODF) algorithm was used to calculate nestedness [56]. Values of nestedness range from 0 to 100, where 0 indicates fully nested networks and 100 represents random networks. Nestedness is a nonrandom pattern where links of specialist species tend to interact with generalist species.

Network indices are not directly comparable given their dependency on network size and therefore were standardized using the Patefield null-model algorithm (r2d method) [57]. The r2d function uses the Patefield algorithm to generate random contingency tables with given marginals while maintaining the sum of the rows and columns. Averages of the null model scores were subtracted from the observed values to obtain the Δw-C, ΔQ, ΔH2′, and Δw-NODF. These values were then divided by the standard deviation of the 1000 null model scores to obtain a Z-score. Z-scores are directly comparable and can also indicate the statistical significance of the network for a particular network index. We considered z-scores with an absolute value greater than 2 to be significant.

#### 2.3.2. Key Plant Species

To quantify which plant species were key in supporting structure and stability of the bee-plant interaction networks, we used the following two approaches: (1) the centrality index approach and (2) the quantitative modularity analysis approach. Both approaches used a quantitative interaction matrix of all bee-plant interactions across all 10 agroforests combined. Bee-plant interaction networks were of the following four types across the 10 agroforests: (1) the rainy season entire bee community, (2) the dry season entire bee community, (3) the rainy season subset community of meliponine bees, and (4) the dry season subset community of meliponine bees.

##### Centrality Index Approach

The centrality index approach follows [58], where the normalized degree, closeness and betweenness centrality measures for all plant species in each network were arcsine transformed and a PCA was conducted. The first principal component was then used as a centrality index for each plant species where positive values indicate a key species that is highly connected and influencing network’s structure and stability and negative values indicate a peripheral species that does not play a significant role [58,59,60].

##### Quantitative Modularity Analysis Approach

The quantitative modularity analysis approach follows Dorman & Strauss [61], and Watts et al. [62] where species are assigned into one of four topological roles in the network. First, the quantitative modularity Q is used to detect modules in the networks, computed using the DIRTLPAwb+ algorithm [63]. Modularity Q indicates the level at which subsets of interactions can be created in a network [61]; values range from 0 to 1, with 0 indicating the network is randomly organized and 1 indicating the network is perfectly modular [62,64]. To account for sampling size and effort, null expectations were calculated using the vaznull method [65] and then used to correct the absolute value of Q by computing the difference between the modularity value of the observed network and the mean modularity value obtained from the 100 null models [2]. The corrected Q was then standardized into a z-score (Z_Q_ = $\frac{Q_{observed}-{\bar{Q}}_{null}}{\sigma_{Q_{null}}}$) [61,64]. This particular z-score was then used to calculate standardized weighted connection and participation coefficients (c- and z-scores) for all plant species in each network, to identify those that held core topological roles in the network [61]. The c-score denotes the level at which a species’ links are equally distributed, that is, its connectivity across modules in the network, where 0 indicates a species’ links mostly within its own module [66,67]. The z-score denotes the intensity of a species’ links or its participation within its own module, e.g., a z-score of 1 indicates that a species’ links are mostly within its own module. Critical thresholds of c- and z-values were established from the null models using 95% quantiles [61,62]. Plant species with c- and z-scores occurring within the defined thresholds are in the periphery of the community or assigned the non-core topological role of peripheral species. Species with c- and/or z-scores exceeding the defined critical thresholds are assigned topological roles as follows: Module Hub (c < c_critical_, z > z_critical_) a non-core topological role; Connector (c > c_critical_, z < z_critical_), a core topological role; and Network Hub (c > c_critical_, z > z_critical_), a core topological role [61].

#### 2.3.3. Bee Community Composition

To evaluate if bee community composition is shaped by agroforestry management practices of diverse agroforests, we conducted a MANOVA and plotted a distance-based redundancy analysis with the ‘capscale’ function in the vegan package [68]. Bee species abundances were quarter power transformed to reduce the effect of variances between species collected with a higher frequency compared to species collected less frequently. The following predictor variables were tested in the analysis: (1) percent shade cover, (2) the number of flowering plant species, (3) the number of tree species, (4) total tree DBH, (5) average tree height, (6) average tree DBH, (7) the total number of open blooms, (8) the total number of trees, and (9) the categorical variable agroforest type. Dissimilarity for MANOVA and distance-based redundancy analysis was based on the Bray-Curtis index.

## 3. Results

### 3.1. Bee Abundance

A total of 3170 bees were aerial netted from flowers during all months of the study. Of these, 1614 were collected in coffee agroforests and a total of 1556 were collected in silvopastural agroforests. Bee abundance did not significantly differ between coffee agroforests and silvopastoral agroforests (Χ^2^ = 0.034; *p* = 0.85, Appendix A).

The variation in bee abundance in diverse agroforests was best described by the model that included only the fixed effect of the number of flowering plant species (22% support). Bee abundance was higher in agroforests and during sampling periods with higher floral species richness (Figure 2A). There were two competing models, the null model with 19% support and a model that included only the fixed effect of the total number of open blooms, with 15% support. However, the 95% CI of the model averaged estimates for the number of flowering plant species and flower abundance span zero, suggesting that there is not enough evidence to conclude that these variables have an effect on bee abundance (Appendix A; Table 2).

### 3.2. Bee Species Richness

A total of 4590 bees representing 5 families and 22 tribes were collected from aerial netting and honey-bait stations. We identified 92% of all specimens to species or morphospecies (Table 3). This included a total of 4243 specimens represented by 4 anthidiine species, 1 apine species, 28 augochlorine species, 9 ceratinine species, 14 megachiline species, 19 meliponine species, and 5 tapinotaspidine species. Only one species was collected from honey-bait stations that was not also netted from flowers, and this species was represented by a single individual and identified as Augochlorini msp 11.

To determine sampling adequacy, a species accumulation curve was constructed and compared to species accumulation curves for the abundance-based estimators ACE and Chao1 (Appendix A). ACE estimated a 90% sampling completeness, and Chao1 estimated an 81% sampling completeness (Appendix A). Rarefaction was also conducted to determine species richness differences between the two agroforestry types. After accounting for abundance differences through rarefaction, a slightly higher number of bee species were found in the silvopasture agroforests (66 species) compared to the coffee agroforests (62 species; Figure 3).

The most plausible model describing bee species richness variation in diverse agroforests was the null model, however the model that included only the fixed effect of the total number of open blooms was a competing model with 14% support (Figure 2B). This relationship was negative, with the number of bee species observed in an agroforest to be higher as the number of open blooms in agroforests and during sampling periods decreased. A third competing model included only the fixed effect of percent shade cover (10% support). Bee species richness was lower in agroforests with greater shade cover. However, the 95% CI of the model averaged estimates for the variables of flower abundance and shade cover span zero, suggesting that there is not enough evidence to conclude that these variables have an effect on bee species richness (Appendix A; Table 2).

### 3.3. Bee Community Composition

The total number of open blooms showed a marginal effect on bee community composition in the diverse agroforests, explaining 17% of the variation (R_2_= 0.17; F_1,8_ = 1.6; *p* = 0.08). We also used the type of agroforest (coffee vs. silvopasture) in the constrained ordination, and while not statistically significant (F_1,8_ = 0.86; *p* = 0.65), it explained 10% of the variation (Appendix A).

### 3.4. Network Metrics

Agroforests differed greatly in all four of the network indices tested: connectance, modularity, specialization, and nestedness (Table 1 and Appendix A).

#### 3.4.1. Connectance

All networks showed a weighted connenctance z-score of <−2.5, indicating that connectance was significantly lower than expected by chance (Table 1). The most plausible model describing variation in network connectance among diverse agroforests for the entire bee community was the model with the fixed effect of the number of tree species (92% support; Table 2). Diverse agroforests with more tree species supported entire bee communities that were less connected (Figure 4A). There were no competing models.

Variation in connectance for meliponine bee networks in diverse agroforests was also best explained by the number of tree species in the agroforest (45% support; Table 2). Diverse agroforests with more tree species supported meliponine bee communities that were less connected (Figure 5A). There was one competing model: the model with the single fixed effect of average tree height (23% support). Diverse agroforests with taller trees supported meliponine bee communities that were less connected.

#### 3.4.2. Modularity

All networks showed a modularity z-score of >2.5, indicating that modularity was significantly greater than expected by chance (Table 1). The most plausible model describing variation in modularity among diverse agroforests for the entire bee community networks was the model with the fixed effect of the number of tree species (49% support; Table 2). Bee-plant networks were more modular in agroforests with higher numbers of tree species (Figure 4B). The only competing model was the null model (23% support).

The most plausible model describing variation in modularity among diverse agroforests for the meliponine bee networks was the null model (26% support; Table 2). There were three competing models: the model with the fixed effect of the number of tree species (19% support); the model with the fixed effect of average tree height (16% support); and the model with the fixed effect of average tree DBH (10% support). All relationships were positive: agroforests with bigger, taller trees, and more tree species supported more modular meliponine bee-plant networks (Figure 5B). However, the 95% CI of the model averaged estimates for these three variables span zero, suggesting that there is not enough evidence to conclude that these variables have an effect on modularity in the meliponine network (Appendix A; Table 2).

#### 3.4.3. Specialization

All networks showed a specialization z-score of >2.5, indicating that specialization was significantly greater than expected by chance (Table 1). The most plausible model describing variation in specialization H2′ among diverse agroforests for the entire bee community networks was the model with the fixed effect of the number of tree species (89% support; Table 2). Agroforests with more tree species hosted more specialized bee-plant networks (Figure 4C). There were no competing models.

The most plausible model describing variation in specialization H2′ among diverse agroforests for the meliponine bee networks was the model with the fixed effect of the number of tree species (42% support; Table 2). Agroforests with more tree species hosted meliponine bee-plant networks that were more specialized (Figure 5C). There were two competing models: the model with the fixed effect of average tree height (19% support); and the null model (18% support).

#### 3.4.4. Nestedness

All networks except one showed a weighted NODF z-score of <−2.5, indicating that nestedness was significantly lower than expected by chance (Table 1). The most plausible model describing variation in nestedness among diverse agroforests for the entire bee community networks was the model with the fixed effects of the number of tree species and the number of flowering plant species (63% support; Table 2). Diverse agroforests with higher numbers of tree species and fewer plant species with open flowers supported entire bee community networks that were less nested (Figure 4D). The only competing model was the model that included only the fixed effect of the number of tree species (29% support).

The most plausible model describing variation in nestedness among diverse agroforests for the meliponine bee networks was the null model (28% support). There were three competing models: the model with the fixed effect of average tree height (22% support); the model with the fixed effect of average tree DBH (15% support); and the model with the fixed effect of the number of flowering plant species (12% support). Networks were more nested in agroforests with smaller trees (Figure 5D). However, the 95% CI of the model averaged estimates for these three fixed effects spans zero, suggesting that there is not enough evidence to conclude that these variables have an effect on modularity in the meliponine network (Appendix A; Table 2).

### 3.5. Key Plant Species

#### 3.5.1. Centrality Index Approach

The centrality index we used combines the species level metrics of normalized degree, betweenness centrality and closeness centrality quantitatively into one value that indicates which species are the most connected within the network.

##### Entire Bee-Plant Network

The rainy season network across all agroforests included 1988 interactions representing 78 bee species and 56 plant species. The five most important flowering plant species during the rainy season were: *Cosmos sulphureus*, *Milleria quinqueflora*, *Lasianthea fruticosa* (all Asteraceae), *Hamelia patens* (Rubiaceae), and *Impatiens balsamina* (Balsaminaceae); (Figure 6; Table 4).

The dry season network across all agroforests included 881 interactions representing 36 bee species and 72 plant species. The top five most important plant species during the dry season were: *Ehretia latifolia* (Boraginaceae), *Triumfetta lappula* (Malvaceae), *Montonoa guatamalensis* (Asteraceae), *Musa acuminata* (Musaceae) and an unidentified white flowering vine species (Figure 6; Table 4).

##### Meliponine Bee-Plant Network

The rainy season network across all agroforests included 1095 interactions representing 19 bee species and 48 plant species. The top five most important flowering plant species during the rainy season were: *Musa acuminata* (Musaceae), *Cosmos sulphureus* and *Milleria quinqueflora* (both Asteraceae)*, Sicyos edulis* (Cucurbitaceae) and *Hamelia patens* (Rubiaceae) (Figure 7; Table 4).

The dry season network across all agroforests included 754 interactions representing 18 bee species and 64 plant species. The top five most important plants species during the dry season were: *Ehretia latifolia* (Boraginaceae)*, Musa acuminata* (Musaceae), *Montonoa guatamalensis* and *Lasianthea fruticosa* (both Asteraceae)*,* and a Piperaceae species (Figure 7; Table 4).

#### 3.5.2. Quantitative Modularity Analysis Approach

The quantitative modularity analysis approach assigns the four topological roles defined in the methods to all species in the network, with peripherals and module hubs considered as non-core topological roles and connectors and network hubs considered as core topological role. Core topological role species are posited to be the most important plant species for the community modeled in the network.

In the rainy season entire bee-plant network no plant species were identified as connectors or network hubs.

In the dry season entire bee-plant network, two Lamiaceae species, *Ocimum basilicum* (basil) and *Salvia colonica* were identified as connectors (Figure 8). No plant species were identified as network hubs.

In the rainy season meliponine bee-plant network, *Byrsonima crassifolia* (Malpighiaceae) was identified as a connector. No plant species were identified as network hubs (Figure 9).

In the dry season meliponine bee-plant network, *Hamelia patens* (Rubiaceae) was identified as a connector. No plant species were identified as network hubs, however, it is worth mentioning that the plant species *Ehretia latifolia* (Boraginaceae) was very close to the cutoff line for network hub (Figure 9).

### 3.6. Key Bee Species

We also used quantitative modularity analysis to identify which bee species are key for supporting network stability and structure. Focusing on the protection of key bee species populations would be an additional potential conservation strategy, i.e., targeting the conservation or increasing abundance of key ecosystem service providers.

In the rainy season entire bee-plant network no bee species were identified as connectors or network hubs (Appendix A).

In the dry season entire bee-plant network, *Trigonisca ziegleri*, *Plebeia jatiformes*, *Melipona panamica* and *Augochlorini* msp. 1 were identified connectors. No bee species were identified as network hubs (Appendix A).

In the rainy season meliponine bee-plant network, *Trigonisca pipioli* and *Plebeia pulchra* were identified as connectors. No bee species were identified as network hubs (Appendix A).

In the dry season meliponine bee-plant network, *Plebeia jatiformes* was identified as connector. No bee species were identified as network hubs (Appendix A).

## 4. Discussion

One of the most important ways that ecological research can support conservation practice is to apply ecological network theory to guide decision-making in land maxing approaches. Here we demonstrate this concept using bee-plant interaction networks. Our results highlight that farmers motivated by their interest in protecting stingless bee communities for their economic value may be able to implement some management actions in diverse agroforests that would also have the unanticipated benefit of supporting the entire bee community. For example, by managing diverse agroforests to ensure the presence of certain plant species that support the stingless bee community, such as *Cosmos sulphereus*, *Milleria quinqueflora*, *Hamelia patens*, *Ehretia latifolia* and *Montonoa guatamalensis*, the entire bee community would also benefit (see Figure 6, Figure 7, Figure 8 and Figure 9).

Our first objective was to understand how bee species richness, bee abundance, bee community composition, and the network metrics of connectance, modularity, specialization and nestedness are related to agroforestry management practices of diverse agroforests. We found that bee species richness, bee abundance, bee community composition and the network indices varied with different management practices, however the number of tree species in an agroforest demonstrated strong relationships with many network indices for both the entire bee network and the meliponine bee network. Total tree abundance ranged from 42–89 individuals in the diverse agroforests we studied, with tree species richness ranging from 18–37 species (Table 1). Agroforests with higher tree species richness were not those with higher tree abundance.

### 4.1. Bee Abundance, Species Richness and Community Composition

Bee abundance did not differ between the two agroforestry systems we evaluated in this study, and bee richness was slightly higher in the silvopasture systems, a result consistent with Galbraith et al. [69], from a slightly lower elevation in Costa Rica. We found that flower abundance and flowering plant species richness were the two most important management practices in diverse agroforests influencing bee abundance, bee species richness and bee community composition. Floral abundance, floral richness and overall plant richness in cultivated lands often have positive relationships with insect pollinator abundance and species richness [70,71], and similarly we found a positive relationship between floral richness and bee abundance. However, we also found that bee species richness was lower in diverse agroforests and during sampling periods with higher numbers of open blooms. The null model was a competing model, with slightly more support, suggesting that the negative relationship between bee richness and floral abundance may be weak and also equally plausible that bee species richness patterns in diverse agroforests are best explained by some other variable we did not measure.

The most important agroforest management variable for bee community composition was the total number of open blooms. Bee species associated with agroforests with higher numbers of open blooms included *Ceratina dimidata*, *Pereirapis semiaurata*, *Plebia pulchra*, *Scaptotrigona subobsucruipennis*, and *Trigona corvina*, whereas bee species associated with diverse agroforests with fewer numbers of open blooms included *Coelioxys* sp. 1, *Geotrigona chiriquiensis*, *Megachile* sp. 13 and *Plebeia jatiformis.*

### 4.2. Connectance

Connectance, or network complexity, decreased with increasing tree species richness in the diverse agroforests we studied, for the entire bee community, as well as for the meliponine bee community. All networks measured were significantly less connected than expected by chance and hyper-diverse tropical communities are posited to have less connected networks due to increased interaction barriers [72] The relationship between connectance and ecological degradation is highly context specific [32] and our study suggests that for tropical bee-plant networks, a negative relationship between connectance and conservation value is supported. Our study also supports a negative relationship between plant species richness and connectance [73,74,75], where connectance can also be viewed as an average of the generalization of the community, and when specialist species are lost, connectance increases.

### 4.3. Modularity

Modularity quantifies compartmentalization in the network, with higher values of modularity indicating more specialization in the network. In this study all tested networks were significantly modular, a finding supported by other studies of tropical lowland bee-plant networks (e.g., [35]). Variation in modularity of bee-plant networks in diverse agroforests was best described by the number of tree species. Modularity was highest in agroforests with more tree species for the entire bee community and for the subset community of meliponine bees. More modular networks imply high niche partitioning or barriers (e.g., spatiotemporal and morphological matching) constraining interactions to a subset of partners [72,76], and more tree species in an agroforest may allow greater niche partitioning for tropical bees.

### 4.4. Specialization

The most important agroforest management practice in diverse agroforests describing variation in network specialization was the number of tree species, both for the entire bee community and the subset community of meliponine bees. In a separate study, a positive association between specialization in hummingbird-plant networks and warmer and more historically stable temperatures found by Martín González et al. [77] was interpreted as showing stronger competition for floral resources in the warmer and more stable conditions of the tropics, where specialization favors species coexistence [78]. The positive association between higher numbers of tree species and specialization found in our study could be interpreted similarly.

### 4.5. Nestedness

The most important agroforestry management practices in diverse agroforests describing variation in network nestedness were the number of tree species for the entire bee community, and the average tree height for the meliponine bee community. Average tree height was also identified as an important agroforest management variable related to connectance for the stingless bee-plant network, where agroforests with taller trees were associated with less connected meliponine bee communities. The majority of stingless bees construct their nests in trees [5], and in the study area only one stingless bee species, *Geotrigona chiriquiensis*, nests in the ground. While conducting our study we searched for stingless bee nests, and asked farmers if they knew of any stingless bee nests on their properties. This led to finding a surprisingly lower number of nests than we had expected, and most of the nests found were located in extremely large *Ficus* spp. trees.

The result that tree species richness is closely associated with network indices for both the entire bee community as well as for the subset of community of meliponine bees may also be partially explained by the high proportion of tree nesting species within the tropical bee community. In the meliponine bee network, all species except one, use trees for nesting, but it is unknown if there are specific tree species that are preferred by each species. In addition, two other tribes included in our entire bee community are also known to use trees for nesting; together totaling 67% of 1988 specimens including all ceratinine, meliponine and apine bees. This number could be higher as the nesting traits for many tropical bee species is incomplete [43,79].

It is crucial to emphasize that, although increased bee abundance and bee species richness are recognized as conservation objectives, additional research is required to identify (1) which network indices are most closely associated with network stability and the provision of ecosystem services [30,33,73]; (2) whether higher or lower values of each network index should be the target of conservation action, e.g., [32]; and (3) whether this is context-dependent, e.g., varies among localities, environmental gradients or network types [72,80]. An additional objective of our study was to contribute datasets to future synthesis efforts aimed at addressing these controversial topics by collecting empirical data from a group underrepresented in the plant-pollinator literature: insects in tropical lowlands [70].

### 4.6. Key Plant Species

The second objective of our study was to quantify which plant species are key in supporting structure and stability of the bee-plant interaction networks, as well as compare these between the entire bee networks and the meliponine bee networks. An important result of this study is a list of quantitatively identified plant species that could be planted by farmers to protect the stability and structure of bee-plant networks in diverse agroforests. Many of the plant species we identified have broad ranges within the neotropics and therefore our results can be applied to diverse agroforestry systems throughout these plant species ranges. However, more empirical bee-plant networks are urgently needed in other lowland tropical ecosystems and regions to develop a more comprehensive list of key plant genera and species. Future studies should also aim to quantify key plant species from bee-plant networks in more natural lowland tropical forests, since more species rich natural tropical ecosystems may function differently compared to the diverse agroforestry systems where our study took place [81], and many more plant species could be included in the potential candidate pool.

Key plant species identified by the two approaches in our study, quantitative modularity analysis and centrality index approach, were more similar than what has been reported in another study comparing these two approaches for fruit-frugivore networks in the study region [60]. The matching/mismatching of key plant species across these approaches as well as other methods we didn’t include here is poorly understood [82] and an area of high research priority to ensure that we are maximizing the potential of restoration and conservation plantings to support biodiversity. In this study we found that key plant species identified via the centrality index approach were identified as module hubs in quantitative modularity analysis, but not as holding either of the two core roles in the network, i.e., connectors or network hubs. In general, module hubs are defined as the most important plant species for one particular module or compartment in the network, not the network as a whole, as a module is defined by species having more interactions within the module than among modules [77]. Currently ecological network theory posits that planting species identified as holding the two core topological roles in fruit-frugivore and plant-pollinator networks is the most effective strategy for ecological restoration or biodiversity conservation initiatives [83,84], however this notion assumes that across different biomes plant species will hold core topological roles in these types of mutualistic networks, but so far the few mutualistic network studies conducted in the tropical lowlands have found that no plant species are both connected and generalist enough to quantitatively be assigned to the core role of network hub [34,35,60]. Our study corroborated this result as we didn’t identify any plant species to hold the core role of network hub in any of the bee-plant networks we constructed. This finding supports our previous suggestion that conservation and restoration plantings for protection of the wild bee community in lowland tropical ecosystems may require the planting of connectors and all module hubs rather than network hubs [35].

Based on our results, we recommend that farmers wishing to protect tropical bee communities in diverse agroforests could plant *Cosmos sulphureus*, *Milleria quinqueflora*, *Hamelia patens* (identified as module hubs & in top five centrality index scores), to support the entire bee community during the rainy season.

In the dry season, tree species *Ehretia latifolia* and *Montonoa guatamalensis* (identified as module hubs & in top five centrality index scores) could be planted to benefit the entire bee community. The key tree species *E. latifolia*, is native throughout Mexico and Central America, and therefore could be broadly planted throughout the region to benefit both the entire bee community and the subset meliponine bee community during its month long bloom during the dry season. For the stingless bee community, farmers could plant *Byrsonima crassifolia*, which was identified as a connector in the dry season meliponine bee-plant network. This tree species is native from Mexico to Bolivia and Brazil, and is also cultivated for its edible fruits known as nance fruits. The Malpighiaceae family, to which this species belongs, has been previously identified as a key plant family for protecting tropical bee-plant networks in Brazil [85].

We also recommend incorporating populations of the plant species *Hamelia patens* which was identified as a key plant species in three out of our four bee-plant networks, twice as a module hub during the rainy season, and once as a connector for meliponine bees during the dry season. This key plant species has a long red, tubular flower that fits the pollination syndrome for hummingbird pollination, yet it is consistently found to be key in supporting bee communities, in this study as well as in a previous study conducted in a very different region of Costa Rica (i.e., the Osa Peninsula; [35]. *H. patens* is a native, weedy, pioneer species colonizing treefall gaps and roadsides from Mexico to Bolivia, and is also planted widely throughout the region as an ornamental.

More empirical bee-plant networks should also be constructed during other times of the year in the tropics, as some plant species identified in our study will only support bee communities during their short bloom times (i.e., <1 mo for *Ehretia latifolia* and *Montonoa guatamalensis*) whereas other plants identified in our study bloom for longer time periods, i.e., *Hamelia patens* has open flowers during all months of the year and *Byrsonima crassifolia* blooms up to three times a year, and each of these three blooms last for approx. 1 month [86]. Other noteworthy key plant species identified in our study that support bees across different seasons owing to their longer duration of flowering phenology include *Lasianthaea fruticosa* and *Musa acuminata*. *L. fruticosa* produces small yellow flowers during most months of the year, ranges from Mexico to Venezuela, and could be planted by farmers to support the entire bee community during the rainy season and the subset meliponine bee community during the dry season. The non-native *M. acuminata* varieties planted in the study area include banana, plantain and cuadrado, all producing slender, tubular white flowers attracting large numbers of stingless bees. This plant species has a long history of cultivation in Central America and is considered naturalized in the region. *M. acuminata* was identified as one of the top five plant species for both the rainy and dry season meliponine bee community, as well as the entire bee community in the rainy season network—however it is important to note that this last result is likely due to its importance for the meliponine bee community and the dominance of meliponine bees in comprising the entire bee community, as 99% of 693 specimens collected in from *M. acuminata* flowers in the study area in 2022 and 2023 were meliponine bees.

### 4.7. Stingless Bees

In Costa Rica there is a growing proclivity towards returning to the traditional practice of stingless beekeeping, and therefore many farmers want to understand how to better manage their lands for stingless bee communities. Stingless bees form eusocial, perennial colonies, workers are active throughout the year [5], and are the dominant members of tropical bee communities. We found that during our study stingless bees comprised 55% of the 1988 bee specimens aerial netted from flowers in the rainy season months, and 86% of the 881 bee specimens aerial netted from flowers during the dry season months. We collected almost the same number of stingless bee species during the dry season (18 species) as we did in the rainy season (19 species), whereas the total number of bee species we collected in the rainy season was more than twice as many as during the dry season (rainy season: 77 species; dry season: 36 species). Most bee species in the tropics are active during the months of peak and post-peak flowering which would run from late dry season (March/April/May) through the early rainy season (June/July) in our study area, although more infrequent, some bee species are only active outside these months [43]. Together these results highlight the importance of considering the seasonality of tropical bee species when developing conservation plans.

Although we did not systematically assess population size over a broad study area, we found several meliponine bee species to be exceptionally rare in the study area suggesting that conservation efforts may need to target these species specifically. For example, the stingless bee species *Cephalotrigona zexmeniae*, *Lestrimelitta danuncia*, *Melipona beechii*, *M. panamica* and *Nannotrigona perilampoides* were only represented by 20 or fewer specimens (Table 3). *M. beechii* and *N. perilampoides* are low elevation species [14], so their low numbers are likely due to their reduced abundances at higher elevations. For the other species, we reviewed the diversity of plant species from which specimens were collected to determine if specialization might be an explanation for the low number of specimens collected. *C. zexmeniae* (20 individuals) was found on 8 different flowering plant species, *M. beechii* (3 individuals) and *M. panamica* (20 individuals) were collected from 3 and 12 different flowering resources, respectively, and, *N. mellaria* (9 individuals) and *N. perilampoides* (3 individuals) were collected from 7 and 2 different flowering plants, respectively. Therefore, it is likely that some factor(s) other than specialization may be responsible for the low population sizes observed in our study. We could not draw any conclusions about the meliponine bee species *L. danuncia* though, as we only collected a single specimen of *L. danuncia*, and this individual was collected from the plant species *Galphimia glauca* (Malpighiaceae). One conservation strategy for these rare bee species may be to use the results of this study to encourage farmers to plant more of the flowering plant species that we collected these bee species from. However, since we did not record whether these floral resources are used as pollen or nectar resources it is challenging to assign importance of the floral resource for the conservation of each species.

### 4.8. Conclusions

It is becoming increasingly important to develop science-based guidelines to maximize the potential of cultivated lands in the tropics to protect ecological process essential for the maintenance of tropical forest biodiversity, and many of these processes are represented by mutualistic networks in ecological network theory, e.g., seed dispersal by fruit-frugivore networks and pollination by plant-pollinator networks. In this paper we demonstrate the application of ecological network science in developing recommendations for farmers to improve their already diverse agroforests for the ecological process of pollination. In addition to providing a list of specific key tree species that farmers can plant to best support bee communities, we also provide evidence that certain management actions such as increasing the number of tree species, and the number of flowering plant species are the best ways to positively impact the bee community. Other agroforestry studies have stressed the importance of increasing wood cover at the landscape scale by conserving natural forest and coordinating agroforestry farms [71]. It is important to reiterate that the agroforests we studied were all embedded in a landscape with high forest cover, and therefore management recommendations from our study can benefit bees as long as the land maxing agroforests are situated in a similar landscape. Future work is needed to apply the objectives of this study to other taxonomic groups that perform ecological services of importance to farmers, such as seed dispersal, pest control and other pollinators, as well as in other regions to develop more comprehensive guides for farmers and land managers.

## Figures and Tables

**Figure 1 insects-16-01269-f001:**
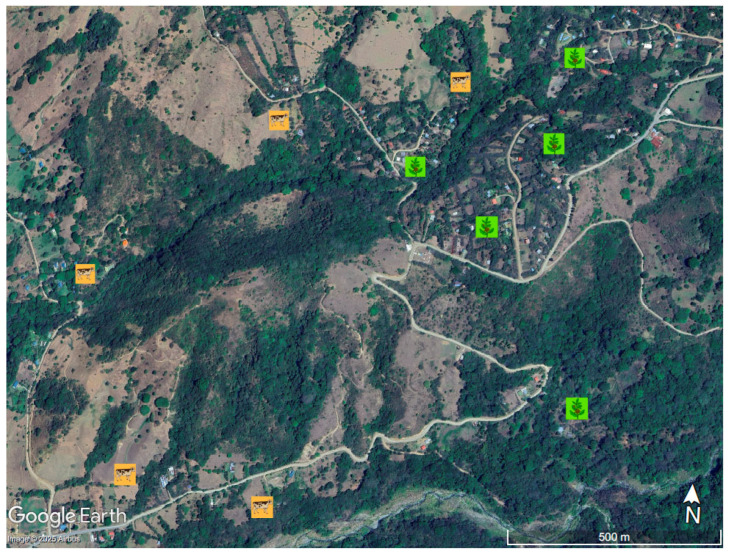
Map of study sites in the San Luis de Monteverde, Puntarenas, Costa Rica. Coffee Agroforestry Systems are denoted with green coffee icon, and Silvopastural Agroforestry Systems are denoted with orange cattle icon. Agroforests were of similar elevation, ranging from 800 to 1000 m.a.s.l. Map center point is located at 10°16′42.73″ N; 84°48′46.45″ W. Map data: Google, Airbus.

**Figure 2 insects-16-01269-f002:**
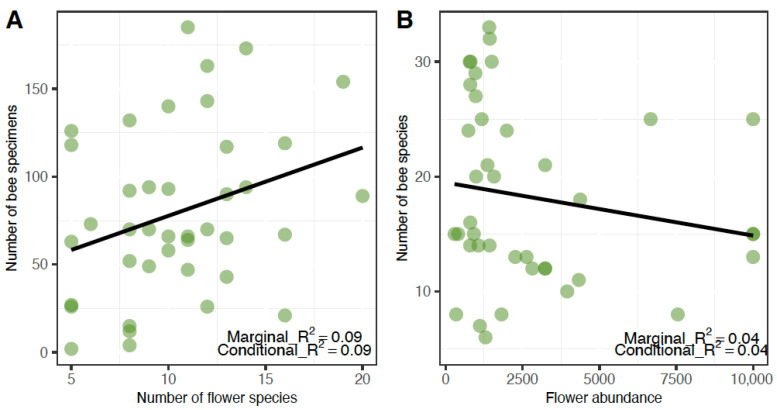
(**A**) Bee abundance and the number of flowering plant species in a diverse agroforest. (**B**) Bee species richness per sampling period and the number of open blooms in a diverse agroforest. Each circle represents one sampling period of an agroforest.

**Figure 3 insects-16-01269-f003:**
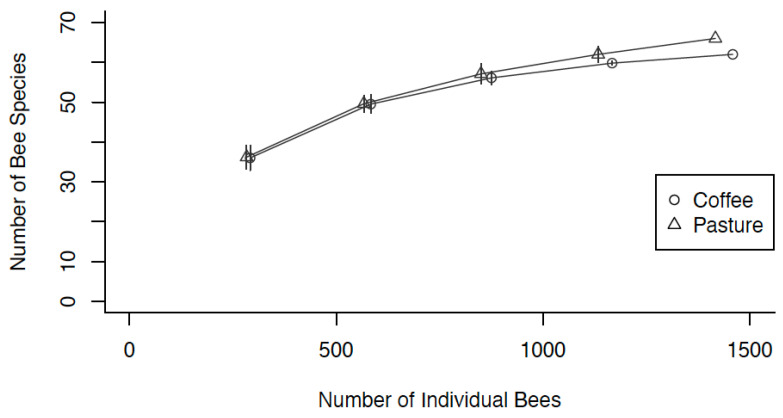
Rarefaction Curve comparing bee species richness between Silvopasture Agroforests and Coffee Agroforests.

**Figure 4 insects-16-01269-f004:**
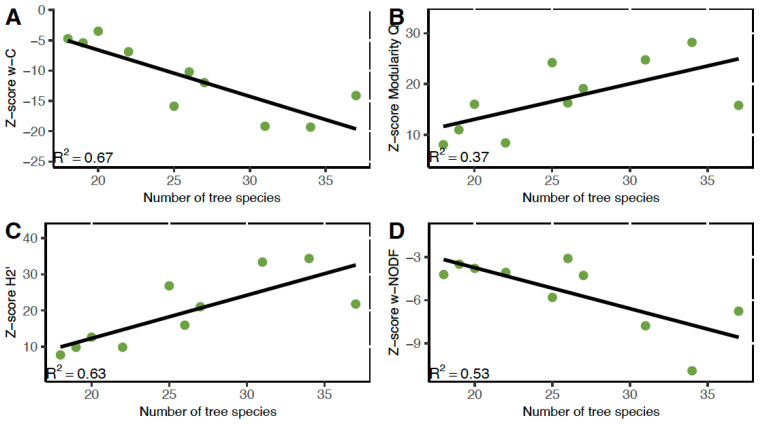
The entire bee community networks most plausible models for (**A**) weighted connectance, (**B**) modularity, (**C**) specialization, and (**D**) nestedness.

**Figure 5 insects-16-01269-f005:**
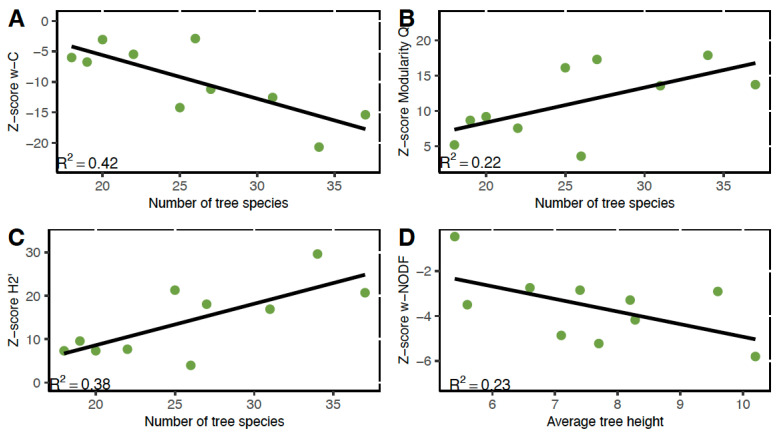
The meliponine bee community networks most plausible models for (**A**) weighted connectance, (**B**) modularity, (**C**) specialization, and (**D**) nestedness. If null model was most plausible model, the relationship of the first competing model is shown.

**Figure 6 insects-16-01269-f006:**
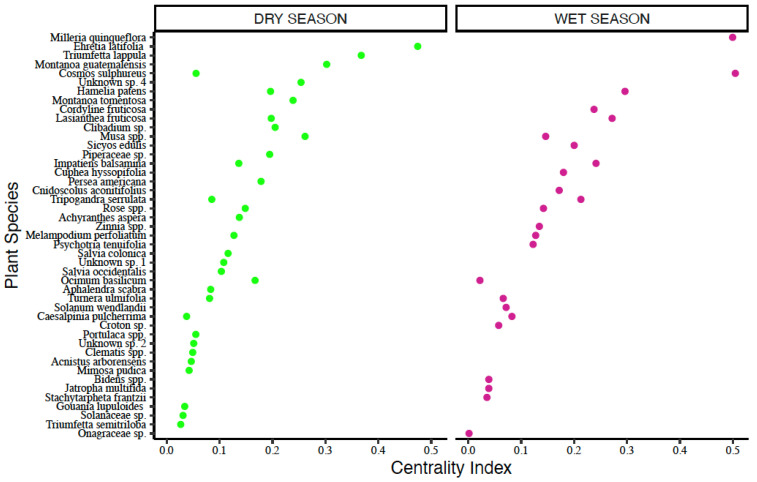
Centrality scores for key plant species in entire bee-plant networks.

**Figure 7 insects-16-01269-f007:**
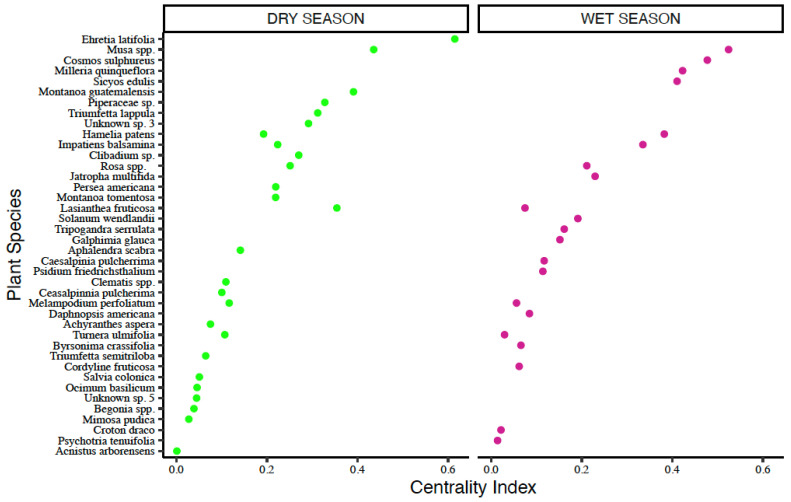
Centrality scores for key plant species in meliponine bee-plant networks.

**Figure 8 insects-16-01269-f008:**
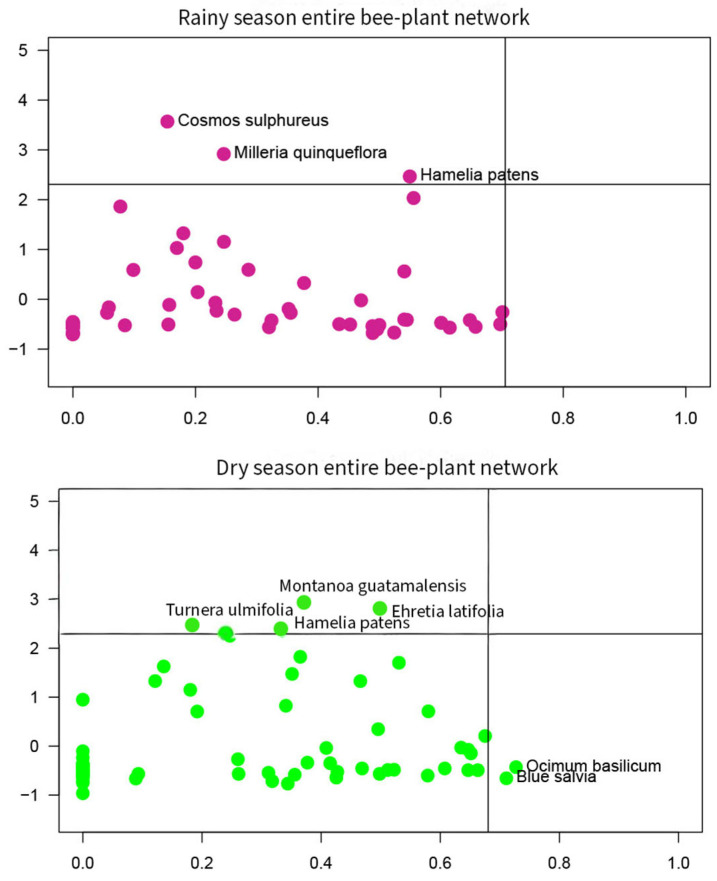
Plant topological roles entire bee-plant networks, rainy and dry seasons.

**Figure 9 insects-16-01269-f009:**
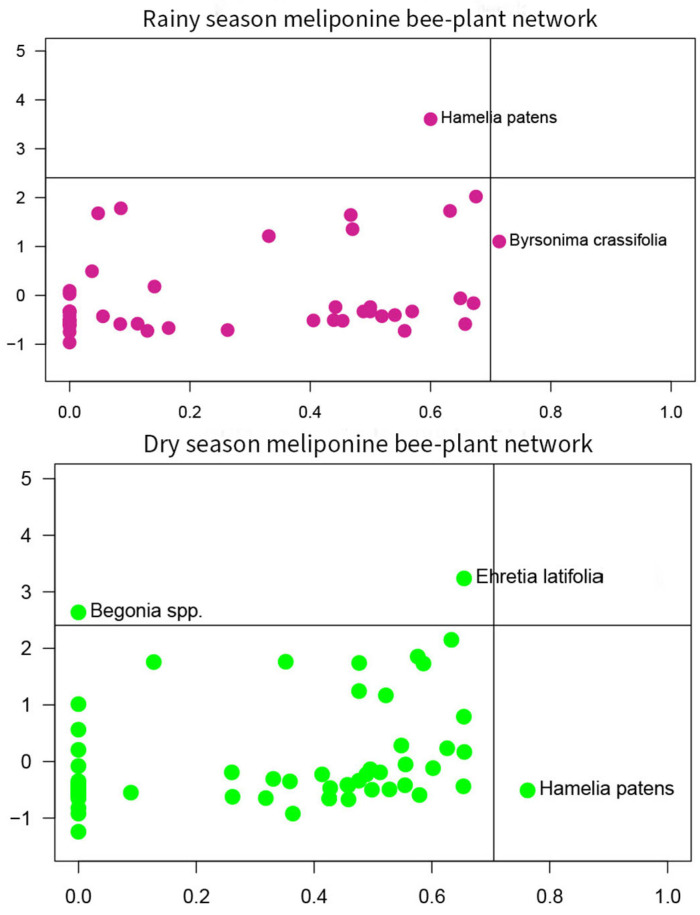
Plant topological roles meliponine bee-plant networks, rainy and dry seasons.

**Table 1 insects-16-01269-t001:** Agroforest characteristics and normalized network metrics, where w-C is weighted connectance, Q is modularity, w-NODF is weighted nestedness metric, and H2’ is specialization.

								Entire Bee-Plant Network						Meliponine Bee-Plant Network					
SiteID	Type	Shade Cover (%)	Ave DBH (cm)	Total DBH (cm)	Ave Tree Height (m)	Tree Richness	Total Trees	Z-w-C	Z-Q	Z-w-NODF	Z-H2′	Plant Species	Bee Species	Z-w-C	Z-Q	Z-w-NODF	Z-H2′	Plant Species	Bee Species
FC	Pasture	62	32.3	1838.4	7.1	31	66	−19.17	24.76	−7.79	33.40	8	27	−12.56	13.56	−4.87	16.93	6	9
GR	Pasture	53	20.6	1629.5	5.4	26	88	−10.21	16.29	−3.09	15.92	16	31	−2.89	3.58	−0.45	3.96	12	7
AL	Pasture	53	28.9	1967.2	7.4	19	67	−5.40	10.98	−3.49	9.77	8	28	−6.74	8.66	−2.85	9.57	7	6
MU	Pasture	32	35.16	2812.4	8.28	18	80	−4.72	8.05	−4.21	7.67	7	29	−6.00	5.19	−4.18	7.32	4	9
RE	Pasture	39	38	1595.7	7.7	25	56	−15.87	24.20	−5.81	26.82	13	39	−14.22	16.11	−5.23	21.32	12	12
HA	Coffee	72	30.2	2654.4	5.6	20	89	−3.49	16.02	−3.78	12.59	13	35	−3.06	9.18	−3.50	7.33	9	8
IA	Coffee	73	19.1	1810	6.6	37	70	−14.12	15.78	−6.77	21.77	12	30	−15.40	13.72	−2.74	20.72	11	10
NS	Coffee	68	33.1	4441.5	10.2	34	80	−19.33	28.19	−10.93	34.39	5	33	−20.70	17.87	−5.81	29.65	5	10
VR	Coffee	67	47.8	1769.8	9.6	27	42	−11.97	19.10	−4.28	21.00	17	28	−11.20	17.29	−2.90	18.08	15	14
WL	Coffee	72	30.6	1622.9	8.2	22	44	−6.88	8.41	−4.06	9.79	10	35	−5.47	7.55	−3.29	7.69	9	7

**Table 2 insects-16-01269-t002:** AICc Model Selection results and model averaged estimates and unconditional S.E.

	Bee Species Richness				Bee Abundance				*w* C				Modularity				H2’				*w* NODF			
Model (Entire)	AICc	ΔAICc	*w*	est ± s.e.	AICc	ΔAICc	*w*	est ± s.e.	AICc	ΔAICc	*w*	est ± s.e.	AICc	ΔAICc	*w*	est ± s.e.	AICc	ΔAICc	*w*	est ± s.e.	AICc	ΔAICc	*w*	est ± s.e.
Null	280.34	0.00	0.26		428.52	0.28	0.19		23.27	8.09	0.02		16.61	1.53	0.23		20.65	6.96	0.03		14.84	6.00	0.03	
Percent Shade Cover	282.26	1.92	0.10	−0.003 ± 0.005	430.99	2.75	0.05	0.001 ± 0.001	27.39	12.21	0.00	−0.005 ± 0.01	20.47	5.39	0.03	0.006 ± 0.01	24.21	10.52	0.00	0.01 ± 0.01	18.61	9.77	0.00	−0.006 ± 0.01
Number of Flowering Plant Species	282.53	2.19	0.09	−0.01 ± 0.02	428.24	0.00	0.22	0.05 ± 0.03	27.3	12.12	0.00	−0.01 ± 0.04	20.40	5.32	0.03	0.02 ± 0.04	24.60	10.91	0.00	0.01 ± 0.04	17.56	8.72	0.01	0.05 ± 0.02
Number of Tree Species	282.81	2.47	0.08	0.001 ± 0.01	430.33	2.09	0.08	0.01 ± 0.02	15.18	0.00	0.92	−0.08± 0.02	15.08	0.00	0.49	0.05 ± 0.02	13.69	0.00	0.89	0.07 ± 0.02	10.38	1.54	0.29	−0.05 ± 0.01
Total Tree DBH	282.71	2.37	0.08	0.07 ± 0.2	430.74	2.50	0.06	−0.18 ± 0.36	27.54	12.36	0.00	0.14 ± 0.66	20.63	5.55	0.03	0.21 ± 0.46	24.86	11.17	0.00	0.13 ± 0.57	16.74	7.90	0.01	−0.56 ± 0.38
Average Tree Height	282.57	2.23	0.09	0.02 ± 0.05	430.75	2.51	0.06	−0.04 ± 0.08	26.34	11.16	0.00	−0.13 ± 0.13	20.70	5.62	0.03	0.04 ± 0.10	24.41	10.72	0.00	0.08 ± 0.12	16.60	7.76	0.01	−0.12 ± 0.08
Average Tree DBH	282.77	2.43	0.08	0.002 ± 0.009	430.72	2.48	0.06	−0.007 ± 0.014	27.49	12.31	0.00	−0.01 ± 0.03	20.56	5.48	0.03	0.01 ± 0.02	24.76	11.07	0.00	0.008 ± 0.02	19.06	10.22	0.00	−0.004 ± 0.02
Total Number of Open Blooms	281.57	1.23	0.14	−0.08 ± 0.07	429.00	0.76	0.15	0.18 ± 0.13	27.39	12.21	0.00	−0.08 ± 0.26	19.39	4.31	0.06	0.20 ± 0.18	24.18	10.49	0.00	0.18 ± 0.22	19.11	10.27	0.00	0.016 ± 0.17
Total Number of Trees	282.82	2.48	0.07	0.001 ± 0.004	430.99	2.75	0.05	−0.001 ± 0.001	26.87	11.69	0.00	0.01 ± 0.01	20.85	5.77	0.03	0.002 ± 0.01	24.86	11.17	0.00	−0.003 ± 0.01	19.12	10.28	0.00	0.001 ± 0.01
Number of Tree Species + Number of Flowering Plant Species	285.11	4.77	0.02		430.40	2.16	0.07		21.04	5.86	0.05		20.68	5.60	0.03		19.45	5.76	0.05		8.84	0.00	0.63	
Model (Meliponine)																								
Null									25.41	2.30	0.14		20.80	0.00	0.26		24.14	1.71	0.18		26.53	0.00	0.28	
Percent Shade Cover									29.62	6.51	0.02	−0.004 ± 0.02	23.71	2.91	0.06	0.013 ± 0.01	27.97	5.54	0.03	0.01 ± 0.02	30.81	4.28	0.03	−0.001 ± 0.02
Number of Flowering Plant Species									29.54	6.43	0.02	0.04 ± 0.06	24.97	4.17	0.03	0.01 ± 0.05	28.40	5.97	0.02	−0.017 ± 0.06	28.26	1.73	0.12	0.099 ± 0.07
Number of Tree Species									23.11	0.00	0.45	−0.07 ± 0.03	21.36	0.56	0.19	0.05 ± 0.02	22.43	0.00	0.42	0.07 ± 0.03	30.77	4.24	0.03	−0.008 ± 0.04
Total Tree DBH									29.31	6.20	0.02	−0.41 ± 0.72	24.84	4.04	0.03	0.26 ± 0.57	27.79	5.36	0.03	0.48 ± 0.67	28.93	2.40	0.08	−0.91 ± 0.70
Average Tree Height									24.46	1.35	0.23	−0.28 ± 0.12	21.78	0.98	0.16	0.19 ± 0.10	23.97	1.54	0.19	0.25 ± 0.12	26.99	0.46	0.22	−0.26 ± 0.14
Average Tree DBH									28.76	5.65	0.03	−0.02 ± 0.03	22.60	1.80	0.10	0.03 ± 0.02	27.00	4.57	0.04	0.03 ± 0.03	27.78	1.25	0.15	−0.04 ± 0.03
Total Number of Open Blooms									29.57	6.46	0.02	−0.09 ± 0.29	23.81	3.01	0.06	0.23 ± 0.22	27.40	4.97	0.03	0.20 ± 0.27	30.61	4.08	0.04	0.12 ± 0.31
Total Number of Trees									28.19	5.08	0.04	0.02 ± 0.01	22.90	2.10	0.09	−0.01 ± 0.01	27.13	4.70	0.04	−0.01 ± 0.01	29.92	3.39	0.05	0.01 ± 0.01
Number of Tree Species + Number of Flowering Plant Species									28.26	5.15	0.03		27.33	6.53	0.01		28.13	5.70	0.02		34.09	7.56	0.01	

**Table 3 insects-16-01269-t003:** Bee species collected and their abundances, msp = morphospecies. Bee tribe and the total bee abundance for each tribe are shown in bold.

Bee Tribe	Bee Species	Aerial Net Flowers		Aerial Net Honey Baits
		Rainy Season	Dry Season	
Anthidiini		**1**	**6**	**0**
	*Anthidiellum msp 1*	1	0	0
	*Anthidiini msp 2*	0	2	0
	*Anthidiini msp 3*	0	3	0
	*Anthidiini msp 4*	0	1	0
Apini		**67**	**60**	**88**
	*Apis mellifera*	67	60	88
Augochlorini		**270**	**23**	**25**
	*Augochlora msp 1*	39	0	2
	*Augochlora msp 11*	1	0	0
	*Augochlora msp 15*	1	0	0
	*Augochlora msp 19*	1	0	0
	*Augochlora msp 20*	1	0	0
	*Augochlora msp 21*	3	0	1
	*Augochlora msp 3*	7	0	0
	*Augochlora msp 5*	17	0	1
	*Augochlora msp 6*	12	3	0
	*Augochlora msp 8*	10	1	4
	*Augochlora nigrocyanea*	8	0	0
	*Augochlorella comis*	16	4	6
	*Augochlorella neglectula*	7	0	6
	*Augochlorini msp 1*	39	0	4
	*Augochlorini msp 10*	1	0	0
	*Augochlorini msp 11*	0	1	1
	*Augochlorini msp 12*	3	2	0
	*Augochlorini msp 2*	2	0	0
	*Augochlorini msp 3*	21	5	0
	*Augochlorini msp 4*	6	0	0
	*Augochlorini msp 6*	4	0	0
	*Augochlorini msp 8*	1	0	0
	*Augochloropsis ignita*	5	0	1
	*Augochloropsis msp 1*	23	1	4
	*Augochloropsis msp 3*	22	6	6
	*Pereirapis semiaurata*	9	0	0
	*Pseudaugochlora graminea*	11	0	0
Ceratinini		**185**	**35**	**5**
	*Ceratina buscki*	37	9	0
	*Ceratina cobaltina*	1	4	1
	*Ceratina dimidata*	11	0	0
	*Ceratina eximia*	24	0	1
	*Ceratina msp. 10*	8	0	1
	*Ceratina msp. 9*	3	0	0
	*Ceratina rectangulifera*	91	17	2
	*Ceratina trimaculata*	7	5	0
	*Ceratina zeteki*	3	0	0
Megachilini		**156**	**0**	**3**
	*Coelioxys msp 1*	13	0	0
	*Coelioxys msp 2*	8	0	1
	*Coelioxys msp 3*	1	0	0
	*Megachile msp 10*	6	0	1
	*Megachile msp 11*	3	0	0
	*Megachile msp 12*	29	0	0
	*Megachile msp 13*	45	0	1
	*Megachile msp 14*	1	0	0
	*Megachile msp 15*	1	0	0
	*Megachile msp 16*	13	0	0
	*Megachile msp 3*	2	0	0
	*Megachile msp 7*	34	0	0
Meliponini		**1095**	**754**	**1252**
	*Cephalotrigona zexmeniae*	6	14	0
	*Geotrigona chiriquiensis*	31	59	23
	*Lestrimelitta danuncia*	1	0	0
	*Melipona beecheii*	1	2	0
	*Melipona panamica*	6	14	2
	*Nannotrigona mellaria*	5	4	50
	*Nannotrigona perilampoides*	1	1	1
	*Partamona orizabaensis*	195	71	433
	*Plebeia frontalis*	28	19	312
	*Plebeia jatiformis*	6	14	62
	*Plebeia pulchra*	31	25	117
	*Scaptotrigona pectoralis*	1	36	0
	*Scaptotrigona subobscuripennis*	11	61	8
	*Tetragona ziegleri*	58	24	3
	*Tetratrigonisca angustula*	76	194	28
	*Trigona corvina*	124	48	44
	*Trigona fulviventris*	312	86	36
	*Trigona silvestriana*	170	58	41
	*Trigonisca pipioli*	32	24	92
Tapinotaspidini		**214**	**3**	**1**
	*Paratetrapedia calcarata*	88	2	1
	*Paratetrapedia connexa*	1	0	0
	*Paratetrapedia lugubris*	42	0	0
	*Paratetrapedia albilabris*	80	1	0
	*Lophopedia pygmaea*	3	0	0
Totals		1988	881	1374

**Table 4 insects-16-01269-t004:** Flowering plant species from which bees were collected. An ‘x’ in the column indicates the species was flowering during the indicated season.

Plant Family	Plant Species	Rainy Season	Dry Season	Top 5 Centrality Index	Topological Role
Acanthaceae	*Aphelandra scabra*		x		
Acanthaceae	*Asystasia gangetica*	x			
Acanthaceae	*Pachystachys lutea*	x			
Acanthaceae	*Sanchezia parvibracteata*		x		
Acanthaceae	*Thunbergia erecta*	x			
Amaranthaceae	*Achyranthes aspera*		x		
Amaranthaceae	*Celosia spicata*		x		
Amaranthaceae	*Gomphrena globoso*		x		
Amaryllidaceae	*Hippeastrum* spp.		x		
Apiaceae	*Eryngium foetidum*		x		
Apocynaceae	*Asclepias curassavica*	x			
Asparagaceae	*Cordyline fruticosa*	x			
Asteraceae	*Acmella radicans*		x		
Asteraceae	*Bidens* spp.	x			
Asteraceae	*Clibadium* sp.		x		
Asteraceae	*Cosmos sulphureus*	x	x	ER; MR	Module hub
Asteraceae	*Lasianthaea fruticosa*	x	x	ER; MD	
Asteraceae	*Melampodium perfoliatum*	x	x		
Asteraceae	*Milleria quinqueflora*	x		ER; MR	Module hub
Asteraceae	*Montanoa guatemalensis*		x	ED; MD	Module hub
Asteraceae	*Montanoa tomentosa*		x		
Asteraceae	*Pseudelephantopus spicatus*		x		
Asteraceae	*Synedrella nodiflora*		x		
Asteraceae	*Tagetes* spp.	x			
Asteraceae	*Zinnia* spp.	x	x		
Balsaminaceae	*Impatiens balsamina*	x	x	ER	
Balsaminaceae	*Impatiens* spp.	x	x		
Begoniaceae	*Begonia* spp.		x		Module hub
Boraginaceae	*Ehretia latifolia*		x	ED; MD	Module hub
Cactaceae	*Cactaceae* sp.	x			
Cleomaceae	*Cleome* spp.		x		
Commelinaceae	*Commelina erecta*		x		
Commelinaceae	*Tripogandra serrulata*	x	x		
Cucurbitaceae	*Cucurbita moschata*		x	MR	
Cucurbitaceae	*Sicyos edulis*	x			
Euphorbiaceae	*Cnidoscolus aconitifolius*	x			
Euphorbiaceae	*Croton draco*	x			
Euphorbiaceae	*Euphorbia graminea*	x			
Euphorbiaceae	*Euphorbia leucocephala*		x		
Euphorbiaceae	*Euphorbia pulcherrima*		x		
Euphorbiaceae	*Jatropha multifida*	x	x		
Euphorbiaceae	*Manihot esculenta*		x		
Fabaceae	*Caesalpinia pulcherrima*	x	x		
Fabaceae	*Inga* spp.		x		
Fabaceae	*Mimosa pudica*	x	x		
Fabaceae	*Senna alata*	x			
Hydrangeaceae	*Hydrangea* spp.	x			
Iridaceae	*Trimezia stayermarkii*		x		
Lamiaceae	*Nepetoideae* sp.		x		
Lamiaceae	*Ocimum basilicum*	x	x		Connector
Lamiaceae	*Pycnanthemum muticum*		x		
Lamiaceae	*Salvia colonica*		x		Connector
Lamiaceae	*Salvia occidentalis*		x		
Lauraceae	*Persea americana*		x		
Liliaceae	*Lilium* spp.	x			
Lythraceae	*Cuphea hyssopifolia*	x	x		
Malpighiaceae	*Byrsonima crassifolia*	x			Connector
Malpighiaceae	*Galphimia glauca*	x			
Malvaceae	*Gossypium* spp.		x		
Malvaceae	*Sida rhombifolia*		x		
Malvaceae	*Triumfetta lappula*		x	ED	
Malvaceae	*Triumfetta semitriloba*		x		
Marcgraviaceae	*Marcgraviaceae* sp.	x			
Melastomataceae	*Pleroma urvilleanum*	x			
Musaceae	*Musa acuminata*	x	x	ED; MR; MD	
Myrtaceae	*Myrtaceae* sp.		x		
Myrtaceae	*Pimenta dioica*		x		
Myrtaceae	*Psidium friedrichsthalium*	x			
Myrtaceae	*Psidium guayaba*	x			
Nyctaginaceae	*Mirabilis jalapa*	x			
Onagraceae	*Ludwigia octovalvis*	x			
Orchidaceae	*Orchidaceae* sp.	x			
Passifloraceae	*Turnera ulmifolia*	x	x		Module hub
Piperaceae	*Piperaceae* sp.		x	MD	
Portulacaceae	*Portulaca* spp.	x	x		
Ranunculaceae	*Clematis* spp.		x		
Rhamnaceae	*Gouania lupuloides*		x		
Rosaceae	*Eriobotrya japonica*	x			
Rosaceae	*Rose* spp.	x	x		
Rubiaceae	*Hamelia patens*	x	x	ER; MR	Module hub; Connector
Rubiaceae	*Pentas* spp.	x	x		
Rubiaceae	*Psychotria pubescens*	x			
Rubiaceae	*Psychotria tenuifolia*	x			
Rutaceae	*Citrus* spp.	x			
Salicaceae	*Casearia sylvestris*		x		
Solanaceae	*Acnistus arborensens*		x		
Solanaceae	*Capsicum* spp.		x		
Solanaceae	*Physalis philadelphica*		x		
Solanaceae	*Solanaceae* sp.		x		
Solanaceae	*Solanum americana*	x			
Solanaceae	*Solanum wendlandii*	x			
Solanaceae	*Streptosolen jamesonii*		x		
Thymelaeaceae	*Daphnopsis americana*	x			
Unknown	Unknown sp. 1		x		
Unknown	Unknown sp. 2		x		
Unknown	Unknown sp. 3		x		
Unknown	Unknown sp. 4		x	ED	
Unknown	Unknown sp. 5		x		
Unknown	Unknown sp. 6		x		
Unknown	Unknown sp. 7		x		
Unknown	Unknown sp. 8		x		
Unknown	Unknown sp. 9	x			
Verbenaceae	*Duranta erecta*	x	x		
Verbenaceae	*Lantana camara*	x	x		
Verbenaceae	*Lantana velutina*	x			
Verbenaceae	*Stachytarpheta frantzii*	x	x		

## Data Availability

The original data presented in the study will be openly available in Symbiota.

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
