# Peer review of "Ecological Network Theory Boosts Land Maxing Benefits for Biodiversity: An Example with Tropical Bee-Plant Interactions"

_insects, 2025, doi:10.3390/insects16121269_

Round 1
Reviewer 1 Report (Previous Reviewer 2)
Comments and Suggestions for Authors
This manuscript reads smoothly and the methods, results, and discussion are well coordinated and together form an integrated whole with clear goals, findings, and conclusions.
I found a few minor items from the revision that need attention and give them below.
Title
Omit period at end
Abstract
- Omit the sentence due to redundancy “The variation in bee abundance in diverse agroforests was best described by the 41 numbers of flowering plant species and was a positive relationship.”
Results
3.3. Bee community Composition
- Paragraph 1, sentence 2: replace “The total number of open blooms” with “it” (too wordy, and clear that it refers to subject of previous sentence).
3.5. Key Plant Species
Meliponine bee-plant network
- Paragraph 1 and 2: for the two species in the Asteraceae, change the wording to :“ Cosmos sulphureus and Milleria quinqueflora (both Asteraceae),” and “Montonoa guatamalensis and Lasianthea fruticosa (both Asteraceae),”
3.5.2. Quantitative modularity analysis approach
- Paragraphs 4 and 5: omit word “Family” in front of plant family names in parentheses
Discussion
4.1. Bee abundance, species richness and community composition
- Last paragraph: this is confusing and needs to be reworded – are you referring to numbers of bee species (richness - this is what you are in fact saying) or are you referring to the abundance of bees, which were especially high in certain species (which I think you intend to say)?
4.4. Specialization
- Paragraph 1: fix wording in second sentence: “In a separate study, a positive association between specialization in hummingbird-plant networks and warmer and more historically stable temperatures found by [75] was interpreted …” You need to either add the author after “by” or change the wording. ALSO, add the “In as separate study” at the start to alert the reader that you are moving to a study other than yours; the current transition is rough.
4.6. Key plant species
- Paragraph 4: fix wording for smoother reading and less repetition: “For the stingless bee community, farmers could plant the Malpighiaceae plant species, Byrsonima crassifolia, which was identified as a connector in the dry season meliponine bee-plant network. This tree species is native from Mexico to Bolivia and Brazil, and is also cultivated for its edible fruits (nance). The Malpighiaceae family, to which this species belongs, has been previously…” Omit “the Malpighiaceae plant species”. What do you mean aby “nance”? Add “to which this species belongs”.
- Paragraph 6: Be consistent: Omit the plant family names in parentheses after the two plant species for which you gave them: (“…phenology include Lasianthaea fruticosa (Asteraceae) and Musa acuminata (Musaceae.”)
Author Response
We sincerely thank the reviewers for their time and the valuable comments and suggestions, which we believe have substantially improved the manuscript. Below, we provide detailed responses to each specific comment. All corresponding changes have been incorporated in the revised version of the manuscript.
Point-by-point response to Comments and Suggestions for Authors
Reviewer #1:
Reviewer #1: This manuscript reads smoothly and the methods, results, and discussion are well coordinated and together form an integrated whole with clear goals, findings, and conclusions.
I found a few minor items from the revision that need attention and give them below.
Comment 1: Title; Omit period at end
Response 1: We removed the period at the end.
Comment 2: Abstract; Omit the sentence due to redundancy “The variation in bee abundance in diverse agroforests was best described by the numbers of flowering plant species and was a positive relationship.”
Response 2: We removed this sentence and improved the result to reflect a more concrete outcome based on a suggestion from Reviewer 2.
Comment 3: 3.3. Bee community Composition; Paragraph 1, sentence 2: replace “The total number of open blooms” with “it” (too wordy, and clear that it refers to subject of previous sentence).
Response 3: We revised this section as suggested and incorporated a suggestion by reviewer 2. Manuscript now states: “The total number of open blooms showed a marginal effect on bee community composition in the diverse agroforests, explaining 17% of the variation (R2= 0.17; F1,8 = 1.6; p=0.08). We also used the type of agroforest (coffee vs. silvopasture) in the constrained ordination, and while not statistically significant (F1,8=0.86; p=0.65), it explained 10% of the variation (Fig. S4).”
Comment 4: 3.5. Key Plant Species; Meliponine bee-plant network; Paragraph 1 and 2: for the two species in the Asteraceae, change the wording to :“ Cosmos sulphureus and Milleria quinqueflora (both Asteraceae),” and “Montonoa guatamalensis and Lasianthea fruticosa (both Asteraceae),”
Response 4: We changed the wording as suggested.
Comment 5: 3.5.2. Quantitative modularity analysis approach; Paragraphs 4 and 5: omit word “Family” in front of plant family names in parentheses
Response 5: We removed “Family” from these paragraphs.
Comment 6: 4.1. Bee abundance, species richness and community composition; Last paragraph: this is confusing and needs to be reworded – are you referring to numbers of bee species (richness - this is what you are in fact saying) or are you referring to the abundance of bees, which were especially high in certain species (which I think you intend to say)?
Response 6: Thank you for this suggestion, we reworded the paragraph to state: “The most important agroforest management variable for bee community composition was the total number of open blooms. Bee species associated with agroforests with higher numbers of open blooms included Ceratina dimidata, Pereirapis semiaurata, Plebia pulchra, Scaptotrigona subobsucruipennis, and Trigona corvina, whereas bee species associated with diverse agroforests with fewer numbers of open blooms included Coelioxys sp. 1, Geotrigona chiriquiensis, Megachile sp. 13. and Plebeia jatiformis.”
Comment 7: 4.4. Specialization; Paragraph 1: fix wording in second sentence: “In a separate study, a positive association between specialization in hummingbird-plant networks and warmer and more historically stable temperatures found by [75] was interpreted …” You need to either add the author after “by” or change the wording. ALSO, add the “In as separate study” at the start to alert the reader that you are moving to a study other than yours; the current transition is rough.
Response 7: We edited this sentence as suggested.
Comment 8: 4.6. Key plant species; Paragraph 4: fix wording for smoother reading and less repetition: “For the stingless bee community, farmers could plant the Malpighiaceae plant species, Byrsonima crassifolia, which was identified as a connector in the dry season meliponine bee-plant network. This tree species is native from Mexico to Bolivia and Brazil, and is also cultivated for its edible fruits (nance). The Malpighiaceae family, to which this species belongs, has been previously…” Omit “the Malpighiaceae plant species”. What do you mean aby “nance”? Add “to which this species belongs”.
Response 8: We revised this section of paragraph 4 as suggested.
Comment 9: Paragraph 6: Be consistent: Omit the plant family names in parentheses after the two plant species for which you gave them: (“…phenology include Lasianthaea fruticosa (Asteraceae) and Musa acuminata (Musaceae.”)
Response 9: We omitted the plant family names from these two places.

Reviewer 2 Report (New Reviewer)
Comments and Suggestions for Authors
The manuscript addresses an interesting and timely topic by linking ecological network theory with biodiversity patterns in agroforests, and the field dataset appears substantial. Overall, the study has the potential to contribute to understanding how management and compositional factors shape pollinator communities and interaction networks, and has clear applied relevance. However, the manuscript requires substantive revision to improve clarity, methodological transparency, and accuracy in the presentation and interpretation of results before it can be considered for publication.
Major concerns:
- The Simple Summary and Abstract are presently too general and do not report concrete outcomes. Key results should be presented quantitatively and unambiguously. For example, instead of stating that bee abundance was “best described by the number of flowering plant species,” report the direction and strength of the relationship (e.g., “bee abundance increased with flowering plant richness, explaining X% of variance; β = …, SE = …, p = …”). Likewise, when claiming that “the number of tree species was the management variable most strongly related to the network indices,” specify which network indices (e.g., connectance, nestedness, modularity) and provide the nature and magnitude of those relationships (effect sizes, confidence intervals or p-values). This will make the manuscript more informative to a broad readership and reduce the risk of overinterpretation.
- Figure 1 appears to reproduce imagery from Google Maps. Because Google Maps content is copyrighted, the authors must either (a) verify and state that they have obtained permission to reproduce the imagery, or (b) replace the figure with a map generated from permissively licensed sources (e.g., OpenStreetMap) and include an explicit source/citation in the caption. The figure caption should also include scale, coordinate reference, and the elevation range or bounding coordinates if relevant to interpretation.
- Insufficient description of sampling design and effort (L170–172; Section 2.2.2). The methods for plant and bee sampling are too brief to ensure reproducibility. It is unclear whether the authors conducted exhaustive surveys within each 1-ha plot, used transects, quadrats, timed searches, or other standard protocols; how many visits/observers per sampling event; and how flowering status was defined and recorded. I recommend (and strongly prefer) a concise table (new) that lists all recorded variables (dependent and independent), their units, whether they are continuous or categorical, and how each was measured or derived (including sampling dates, number of visits, observer effort, and any conversion formulas). This table will greatly clarify subsequent analyses and facilitate replication.
- L184: The term “aerial nets” is ambiguous in an entomological context and can be misinterpreted (e.g., nets for birds or bats, or remote-sensing connotations). Please replace with precise terminology—e.g., “entomological sweep nets” or “hand-held insect nets (standard entomological nets)”—and, if relevant, provide mesh size and sampling protocol (sweep length, duration, or number of strokes).
- Tree measurements and DBH tape (L221). The methods state that DBH was measured “using a DBH tape.” Because a DBH tape records trunk circumference, which is then converted to diameter by the tape’s scale, the authors should clarify whether circumference was recorded and converted to diameter (and report the formula used), or whether a different device was used to measure diameter directly. This distinction affects data accuracy and should be explicit.
- Species accumulation / sampling completeness (L399–400). The text states that the species accumulation curve shows sampling was “nearly sufficient,” yet no quantitative completeness metrics are provided. First, ensure figure references are correct: the text cites Fig. S3 but the curve appears to be in the main manuscript; update the reference for consistency. Second, provide exact metrics of completeness (for example, estimated total species from Chao1 or Jackknife, and the ratio observed/estimated or percent completeness). Reporting those numbers will allow readers to assess representativeness objectively rather than relying solely on a visual impression.
- Statistical interpretation: MANOVA result (L417–419). The manuscript asserts that “the only agroforest variable identified by MANOVA to influence bee community composition was the total number of open blooms (p = 0.08).” A p-value of 0.08 does not meet the conventional α = 0.05 threshold for significance; therefore, it is inaccurate to state that the variable “was identified … to influence” community composition. The authors should correct the wording to reflect the result appropriately (for example, “the total number of open blooms showed a marginal effect on community composition (MANOVA, p = 0.08)”), or present effect sizes and confidence intervals and discuss the result as non-significant but potentially suggestive. Avoid presenting marginal or non-significant results as significant.
Minor points and stylistic issues:
- Check all figure and table references for consistency between the main and supplementary materials. Several in-text references to supplementary figures appear to be misplaced or outdated.
- Consider adding brief methodological references or citations for standard protocols used (e.g., bee sampling methods, network metrics calculation) so readers unfamiliar with the protocols can consult primary sources.
- When discussing network indices, consider reporting both raw and normalized values where appropriate, and give readers a sense of biological relevance (i.e., what does a change of X units mean ecologically).
Author Response
We sincerely thank the reviewers for their time and the valuable comments and suggestions, which we believe have substantially improved the manuscript. Below, we provide detailed responses to each specific comment. All corresponding changes have been incorporated in the revised version of the manuscript.
Reviewer #2:
Reviewer #2: The manuscript addresses an interesting and timely topic by linking ecological network theory with biodiversity patterns in agroforests, and the field dataset appears substantial. Overall, the study has the potential to contribute to understanding how management and compositional factors shape pollinator communities and interaction networks, and has clear applied relevance. However, the manuscript requires substantive revision to improve clarity, methodological transparency, and accuracy in the presentation and interpretation of results before it can be considered for publication.
Major concerns:
Comment 1: The Simple Summary and Abstract are presently too general and do not report concrete outcomes. Key results should be presented quantitatively and unambiguously. For example, instead of stating that bee abundance was “best described by the number of flowering plant species,” report the direction and strength of the relationship (e.g., “bee abundance increased with flowering plant richness, explaining X% of variance; β = …, SE = …, p = …”). Likewise, when claiming that “the number of tree species was the management variable most strongly related to the network indices,” specify which network indices (e.g., connectance, nestedness, modularity) and provide the nature and magnitude of those relationships (effect sizes, confidence intervals or p-values). This will make the manuscript more informative to a broad readership and reduce the risk of overinterpretation.
Response 1: We appreciate this suggestion and have revised the key results of the abstract to report more concrete outcomes as suggested. Specifically, we revised the key results section of the abstract to provide percent variance explained or R2 values, as well as coefficients and SE. We avoided p-values as they are generally not suggested to present these when using an AIC model selection approach.
Key results section now states “Bee abundance increased with flowering plant richness, explaining 9% of the variance (R2=0.09; β = 0.05, SE = 0.03). Diverse agroforests with higher numbers of tree species supported less connected (R2=0.67; β = -0.08, SE = 0.02), less nested (R2=0.53; β = -0.05, SE = 0.01), and more specialized (R2=0.63; β = 0.07, SE = 0.02) and modular (R2=0.37; β = 0.05, SE = 0.02) bee-plant networks.”
Comment 2: Figure 1 appears to reproduce imagery from Google Maps. Because Google Maps content is copyrighted, the authors must either (a) verify and state that they have obtained permission to reproduce the imagery, or (b) replace the figure with a map generated from permissively licensed sources (e.g., OpenStreetMap) and include an explicit source/citation in the caption. The figure caption should also include scale, coordinate reference, and the elevation range or bounding coordinates if relevant to interpretation.
Response 2: According to the website: https://about.google/brand-resource-center/products-and-services/geo-guidelines/#required-attribution “Google Earth or Earth Studio can be used for purposes such as research, education, film and nonprofit use without needing permission. All content created from Google Earth or Earth Studio must always be properly attributed. You can find the attribution in the line(s) shown on the bottom of the content in our mapping products along with copyright notices, such as “Map data ©2019 Google”. Note that the exact text of the attribution changes based on geography and content type.”
Following these guidelines, we updated our Figure 1 map in Google Earth Pro to be sure to include the most recent way that Google Earth labels the attribution. In our previous map the airbus data was shown but not with the current copyright symbol. We also updated the figure caption to include the correct way to attribute the data according to the official guidelines from google website above. Scale is included on map, and figure caption was updated to include a coordinate reference and the elevation range.
Comment 3: Insufficient description of sampling design and effort (L170–172; Section 2.2.2). The methods for plant and bee sampling are too brief to ensure reproducibility. It is unclear whether the authors conducted exhaustive surveys within each 1-ha plot, used transects, quadrats, timed searches, or other standard protocols; how many visits/observers per sampling event; and how flowering status was defined and recorded. I recommend (and strongly prefer) a concise table (new) that lists all recorded variables (dependent and independent), their units, whether they are continuous or categorical, and how each was measured or derived (including sampling dates, number of visits, observer effort, and any conversion formulas). This table will greatly clarify subsequent analyses and facilitate replication.
Response 3: We revised the entire methods section to improve clarity and details of the methods, specifically addressing the concerns listed by the reviewer above, including that exhaustive surveys were conducted within each 1-ha plot and that within these exhaustive surveys, each plant species with open flowers was observed for bee visits to flowers. We also constructed a table that included all the suggested elements, as supplemental Table S1.
Comment 4: L184: The term “aerial nets” is ambiguous in an entomological context and can be misinterpreted (e.g., nets for birds or bats, or remote-sensing connotations). Please replace with precise terminology—e.g., “entomological sweep nets” or “hand-held insect nets (standard entomological nets)”—and, if relevant, provide mesh size and sampling protocol (sweep length, duration, or number of strokes).
Response 4: Thank you for this suggestion, we revised the manuscript to state “Once a bee was observed to land on a flower, a hand-held insect net (Lito Enterprise Society) or plastic collecting jar (Bioquip) was used to target the collection of that individual. Only one attempt was made to collect each bee that landed on a flower, as all missed attempts would result in the bee flying away. If a plant species’ open flowers were all above 8m, bee sampling was not conducted, although this only occurred for one tree species in one of the agroforests.”
Comment 5: Tree measurements and DBH tape (L221). The methods state that DBH was measured “using a DBH tape.” Because a DBH tape records trunk circumference, which is then converted to diameter by the tape’s scale, the authors should clarify whether circumference was recorded and converted to diameter (and report the formula used), or whether a different device was used to measure diameter directly. This distinction affects data accuracy and should be explicit.
Response 5: To address this concern we clarified in the manuscript: “Tree size was determined by two measurements: 1) the trunk diameter at breast height (DBH) using a DBH tape, and 2) tree height, using a rangefinder. We used the standard geometric formula of diameter= circumference/ that is calibrated on the DBH tape itself.”
Comment 6: Species accumulation / sampling completeness (L399–400). The text states that the species accumulation curve shows sampling was “nearly sufficient,” yet no quantitative completeness metrics are provided. First, ensure figure references are correct: the text cites Fig. S3 but the curve appears to be in the main manuscript; update the reference for consistency. Second, provide exact metrics of completeness (for example, estimated total species from Chao1 or Jackknife, and the ratio observed/estimated or percent completeness). Reporting those numbers will allow readers to assess representativeness objectively rather than relying solely on a visual impression.
Response 6: We reviewed the figure references, and the figure in the main manuscript, Fig.3, shows a rarefaction comparison of species richness between agroforestry types, while the Fig. S3 in supplemental material was the accumulation curve showing sampling completeness. Thus we retained the figure references as they were in the text.
We updated our accumulation curve in the supplemental text, Fig S3, to include the accumulation curve of our sampled species richness as well as two abundance-based estimators of sampling completeness, Chao1 and ACE. We also updated the manuscript with the estimated total species from the Chao1 and ACE estimates and calculated the percent completeness using these metrics.
We added the lines: “To determine sampling adequacy, a species accumulation curve was constructed and compared to species accumulation curves for the abundance-based estimators ACE and Chao1 (Fig. S3). ACE estimated a 90% sampling completeness, and Chao1 estimated an 81% sampling completeness (Fig. S3).”
Comment 7: Statistical interpretation: MANOVA result (L417–419). The manuscript asserts that “the only agroforest variable identified by MANOVA to influence bee community composition was the total number of open blooms (p = 0.08).” A p-value of 0.08 does not meet the conventional α = 0.05 threshold for significance; therefore, it is inaccurate to state that the variable “was identified … to influence” community composition. The authors should correct the wording to reflect the result appropriately (for example, “the total number of open blooms showed a marginal effect on community composition (MANOVA, p = 0.08)”), or present effect sizes and confidence intervals and discuss the result as non-significant but potentially suggestive. Avoid presenting marginal or non-significant results as significant.
Response 7: Thank you for making this important point. We revised the manuscript to state:”The total number of open blooms showed a marginal effect on bee community composition in the diverse agroforests (R2=0.17; F1,8 = 1.6; p=0.08).”
Minor points and stylistic issues:
Comment 8: Check all figure and table references for consistency between the main and supplementary materials. Several in-text references to supplementary figures appear to be misplaced or outdated.
Response 8: We reviewed all figure and table references for consistency between the main and supplementary materials and didn’t find any discrepancies.
Comment 9: Consider adding brief methodological references or citations for standard protocols used (e.g., bee sampling methods, network metrics calculation) so readers unfamiliar with the protocols can consult primary sources.
Response 9: We thank the reviewer for this comment. Currently we have method references for each network metric calculation in the methods section, but we added references for the honey bait and netting methods for bee sampling.
Comment 10: When discussing network indices, consider reporting both raw and normalized values where appropriate, and give readers a sense of biological relevance (i.e., what does a change of X units mean ecologically).
Response 10: We appreciate this comment and agree it would be important to be able to provide a biological relevance for the network indices. Owing to the debate about which network indices relate best to conservation value and whether higher or lower values of the index is related to a higher conservation value, we intentionally avoided an ecological interpretation of the network indices in the manuscript. We added a supplemental table (Table S2) of the raw values of each network index in the case that future studies would like to use them for comparison to values they obtain.
Round 2
Reviewer 2 Report (New Reviewer)
Comments and Suggestions for Authors
The authors have addressed all comments from the previous round. I have no further suggestions. The manuscript is suitable for acceptance.
This manuscript is a resubmission of an earlier submission. The following is a list of the peer review reports and author responses from that submission.
Round 1
Reviewer 1 Report
Comments and Suggestions for Authors
Dear Authors,
Your study proposal is very interesting and necessary, although it involves great responsibility. Therefore, you need to be rigorous in your methodology, in data collection and, mainly, in the number of agroflores sampled. So I think this number should be at least double what you used. It is also very important to say the distance between them, which should be at least 1 km. I also think that perhaps it would be more informative if you used landscape metrics, as these agroforests are inserted in a landscape context, and are certainly influenced by the surroundings. Regarding the analyses used, I think you need to be very careful when using too many models as they can obscure the results and you may not be able to understand what actually occurs in nature. For this reason, we often end up going around in circles in the discussion and not discussing what the study's proposal really is. I think this happened to you, because you didn't come up with the proposal for the study title. I suggest that you rethink your data and focus the discussion on what was actually proposed. And be careful when suggesting strategies directly to rural producers so that they do not generate an even greater environmental impact. I made several comments throughout the text. Good luck.

I think English needs to be revised and written in a more scientific format. Many paragraphs are long and sentences in the same paragraph start the same way.
Reviewer 2 Report
Comments and Suggestions for Authors
Please see the attached word document.

Round 2
Reviewer 2 Report
Comments and Suggestions for Authors
Review of second submission of “Ecological network theory to guide decision-making in land maxing approaches: an example with tropical bee-plant networks” – Insects – 3731194
Overall, the manuscript reads much more smoothly and clearly. The authors addressed all of the comments to improve the paper and I have only a few minor comments on this resubmission.
Materials and methods
Data collection
- In opening paragraph, do not repeat the year and add a space between the date and the month. “ The entire… bees in 2022, three times… season, 5 June - 24 July, and once… season, 7 - 19 December.”
Bee sampling
- First paragraph, near end :
- make timed observations clearer and omit repetition of flowering for plants that have flowers: “…for at least four 30-minute observations during… that the plant species had open flowers.”
- Last sentence: write out “1” as “one”
- Second paragraph: avoid repeating entire “honey-bait station” when this is clear from the context, AND make same changes as above for the timed observations:
- “Honey-bait stations were comprised of… with one station located close… and a second located near..”
- “Bees were collected… each station for two 30-minute observations.. and the second between 12:00-14:00 hours.”
- Last sentence: replace “while” with “whereas”, which is the preposition for making contrasts
Plant sampling
- First paragraph, middle sentence: “ In addition,… each individual plant was…” [a plant with blooms is a flowering plant! Avoid wordy repetitions]
Network metrics
- First paragraph
- Omit new sentence (in green, “Each…agroforest”); it adds no new or important information, and is confusing
- The following sentences are confusing and do not make sense; I suggest:
Make one single sentence “Bee-plant… meliponine bees.” Follow this with “Both network types included the individual agroforest interactions across all three … rainy season”. Follow this with “Dry…due to the small sample size of only one sampling period.” Then, “Four network metrics were selected: 1) weighted connectance to represent connectance, 2) modularity Q to determine modularity, 3) H2 to measure specialization , and 4) weighted NODF [give the full name of this along with the acronym] to show nestedness.” This makes it clearer for the reader.
Results
Bee abundance
- First paragraph, simply the sentence with the bee specimens numbers: “A total of…study; of these, 1,614 were… and 1,556 in silvopastural agroforests.”
Key Plant Species
Throughout this section:
- Organize this more clearly and be more concise and less repetitive by adding subheadings “Entire bee-plant network” and “Meliponine bee-plant network” in each section, and then under each addressing, as you do already, the rainy season in one paragraph and the dry season in a second paragraph. Be sure to omit all “entire bee-plant network” and “meliponine bee-plant network” within each paragraph – no need to have this when you add a subheading.
- When you give a family name in parentheses, omit the word “family” – it is obvious by the name, which has a family-specific ending! [this applies to all taxonomic groups]
- For example:
- Centrality index approach, second paragraph: “The rainy season across… Cosmos sulphureus (Asteraceae), Milleria quinqueflora (Asteraceae), etc…”
- To be even more structured, list the species that are in the same family next to each other and, instead of putting the same family name after each one, put “ (all Asteraceae)” after the last species in that shared family. This also helps the reader see patterns in the family representation of the species.
- Figure 6 and Figure 7: use a font for the plant species names that is sharper and legible! Consider making the names larger. Also, be sure that the font is in black. If the reader cannot read the species names, the entire figure is worthless!
Important Bee species – the numbering of this section is off (is it supposed to be 3.5.3?) Be sure you have numbered all of your sections properly
Discussion
4.1, second paragraph:
- What is the reason behind the order in which the bees are listed? It seems that having them in alphabetical order would be clearer for the reader, or are you listing them by importance? If by importance, you need to say so.
4.2 : When listing two or more references together, list from oldest to most recent.
4.6, second paragraph
- Second sentence does not make sense and is not smoothly integrated with the previous one – what are the “such” cases of overlap between these approaches? Make clear that you are referring to YOUR data. You may want to rethink the organization of this paragraph.
Third paragraph:
- The lead sentence is confusing… Maybe just start with “Based on our results, we recommend..”
- This paragraph is confusing because of the jumbled discussions that refer to rainy and dry seasons. Put your recommendations and their discussion for the wet season in one paragraph, and then your recommendations for the dry season in a separate paragraph – do not mix them up.
- In each paragraph, once you give the full plant species (binomial), abbreviate the genus when giving the name again. **This applies to other paragraphs as well!
Stingless bees
- Second paragraph: In sentence listing the bee species, give the full genus of each species except if it follows a congeneric species (such as the case for panamica). For the rest of the paragraph, abbreviate the genus for each species in a consistent manner.
Conclusion
- Last sentence: “taxonomic groups” is very broad and vague– maybe follow by an example “ such as…”? Give a sense of broadly you thinking here in terms of network studies for the future.
Tables
- Table 1:
- How are the sites ordered? It seems that all the pasture should be together, and then all the coffee? This comparison is focus of the study. I would also remove the landowner name, and instead just have the initial(s) of their name as the id of the site.
- Put a line under each network heading that extends above all the columns that belong to each network to make clear where the data fit
- Tree height should come after the DBH columns – organize the columns logically.
- Tables 2 and 3 are good. Define msp in the caption (msp = morphospecies).
- Family names are not italicized (Cactaceae sp.), and the abbreviations “sp” (= one species) and “spp” ( = more than one species) are not italicized
